# Low-Carbon Travel Behavior in Daily Residence and Tourism Destination: Based on TPB-ABC Integrated Model

**Liying Wang [1]** **, Junya Wang [2], Pengxia Shen [1], Shangqing Liu [1] and Shuwei Zhang [1],***

[1]  School of Economics and Management, Beijing Jiaotong University, Beijing 100044, China; wangliying@bjtu.edu.cn (L.W.); 19113081@bjtu.edu.cn (P.S.); 21113086@bjtu.edu.cn (S.L.)

[2]  Faculty of Architecture and City Planning, Kunming University of Science and Technology, Kunming 650504, China; wangjunyakust@126.com

*  Correspondence: 17113183@bjtu.edu.cn; Tel.: +86-133-5050-7872

**Abstract:** Low-carbon travel is considered as one of the most important strategies to reduce transportation carbon emissions, and its success is decided by the active participation of residents. Based on the theory of planned behavior (TPB) and Attitude-Behavior-Context theory (ABC), this study explores the influencing factors and formation paths of individual low-carbon travel behavior, and analyzes low-carbon travel behavior regarding both daily commuting from residence and tourism destinations. This study collects a sample of 506 respondents and uses Mplus 8.0 to examine the hypotheses. Empirical research results indicate that: (1) A certain gap exists in the individuals' low-carbon travel behavior between daily residence and tourism destination. Differences exist in direct effects, mediating effects and moderating effects. (2) Low-carbon travel behavioral intention plays a significant mediating role in both daily residence and tourism destination, especially the former. Regarding daily residence, individuals' attitude, subjective norms and perceived behavioral control have a positive effect on behavior through behavioral intention. Regarding tourism destination, only the attitude-low-carbon travel behavioral intention-behavior path is significant. (3) Situational factors play a significant positive moderating effect on the relationship between low-carbon travel behavioral intention and behavior, especially in tourism destination. This study reveals the internal mechanism of individuals' low-carbon travel behavior and the differences between travel in daily life and tourism, helping to deepen understanding of individuals' low-carbon travel behavior and providing guidance for promoting individuals' low-carbon travel.

**Keywords:** low-carbon travel behavior; the theory of planned behavior; Attitude-Behavior-Context theory; daily residence; tourism destination

## 1. Introduction

Carbon emissions from the transportation sector are increasingly contributing significantly to overall carbon emissions due to the growing use of coal, increasing pas-senger travel demand, and longer travel distances [1–5]. Several nations and regions have implemented various measures to tackle these issues [6]. For example, cities such as Beijing, Shanghai, and Guangzhou have implemented policies restricting the movement of vehicles based on license plate numbers [7]. In addition, a typical measure to reduce the number of cars is the license plate lottery system, which is currently in place in cities like Beijing, Tian-jin, Shenzhen, and Hangzhou [8,9]. However, solely focusing on legislation is insufficient for long-term improvements in transportation carbon emissions [10]. It is critical to develop cleaner transportation modes. In light of these circumstances, low-carbon transportation has increasingly becoming a global trend [11]. Making individual behavior adjustments can significantly reduce greenhouse gas emissions [12,13]. The low-carbon travel behavior (LTB) of passengers is a fundamental factor that influences the expansion of low-carbon transportation [14]. Currently, promoting the shift of individuals from high-carbon travel to

low-carbon travel represents a significant challenge [15,16]. Consequently, it is imperative to further examine the factors and pathways that influence individual's LTB.

Several recent studies have demonstrated variations in individuals' pro-environmental behavior across different contexts, including family and workplace environments [12,17], and family and holidays [18]. Previous studies highlighted that, despite individuals' generally high awareness of pro-environmental issues, their pro-environmental behavior tends to decrease when transitioning from a home environment to a holiday setting [18]. Individuals who exhibit higher environmental consciousness at home tend to utilize fewer environmental transportation modes while traveling [6,19]. While previous studies have examined the differences in individuals' travel behavior between daily and tourism contexts, the majority of research has primarily focused on travel behavior between cities rather than within-destination [20,21]. Travel behavior between cities is heavily influenced by the availability of transportation modes [22,23]. Furthermore, previous studies have employed multiple samples to investigate variations in travel behavior between local and tourism destinations [24], but this approach has resulted in limitations and hindered the ability to conduct meaningful comparative analyses across different contexts [25]. This study employs a single sample to investigate the individuals' LTB in both daily residence and tourism destinations, thereby bridging the gaps in existing research.

The theory of planned behavior (TPB) describes individual behavior and has been extensively utilized in studying pro-environmental behavior [26–28] and LTB [4,29,30]. According to TPB, an individual's behavioral intention is determined by attitudes, subjective norms, and perceived behavioral controls, and behavior is determined by behavioral intentions [31]. Attitude refers to an individual's positive or negative attitude towards a certain behavior, subjective norm refers to the social pressure that individuals perceive to participate or not in a certain behavior, and perceived behavioral control refers to the degree of difficulty an individual perceives for a certain behavior [31]. Intention is not the sole determinant of behavior according to major behavioral theories, and the model needs to be expanded by incorporating additional factors [32]. Numerous studies have extended the TPB model to assess individuals' behavior [7,33–35].

Although TPB considers the influence of individual factors on behavior, it ignores the external influence of contextual factors [36]. Attitude-Behavior-Context (ABC) theory highlights the significance of context in shaping individuals' environmental behavior [37]. Individual behavior is strongly influenced by situational factors [38]. Failure to consider contextual factors hinders the accurate prediction of behavior [39]. ABC states that "A" denotes an individual's attitude toward a certain conduct (including attitude, intention, and emotion), "B" denotes a particular behavior, and "C" denotes contextual variables [37]. Studies have shown that contextual factors are beneficial in mitigating the gap between intention and behavior [40]. Currently, some studies integrate TPB with ABC theory to predict environmental behavior. For instance, they employ the TPB-ABC integrated model to examine residents' waste sorting behavior [41] and sustainable consumption behavior [42].

This study, based on the TPB-ABC integrated model, aims to examine the determinants influencing individuals' LTB and explore the differences in LTB between daily residence and tourist destinations. The study has the following specific objectives: (1) investigate the main factors that influence LTB by combining TPB and ABC theory; (2) examine the moderating effect of situational factor (SF); (3) explore the gap between LTB in daily residence and tourism destination. In addition, this paper makes three main contributions. First, the TPB-ABC integrated model is used to explore the individuals' LTB, which is an expansion of TPB theory and ABC theory. Second, this study uses the TPB-ABC integrated model to explore the influencing factors and formation paths of individuals' LTB, which is conducive to deepening the understanding of LTB and further deepens the existing research on LTB. Third, this study distinguishes the differences between LTB in different contexts. Previous studies distinguished differences between individuals' travel behavior in home and tourist destinations [21,25,43], but focused on travel behavior between cities rather than within

cities [21]. In addition, a study has used multiple samples to explore the differences in individual LTB in different contexts [24], which was a limitation of the research. This study bridges the gaps in existing research and deepens research on LTB in different contexts.

The structure of this paper is as follows. Section 2 expounds the theoretical background and research framework and puts forward the research hypotheses. Section 3 introduces the research object, data collection process, measurement scale and research method. Section 4 explains the research results, including the descriptive statistical analysis, reliability and validity analysis, and hypotheses test. The last part reports the discussion and conclusion.

## 2. Literature Review and Research Hypothesis

### 2.1. LTB

Low-carbon behavior is defined as behavior that enhances the resource efficiency of energy or materials, consequently contributing to the dynamics and structure of ecosystems or biospheres [39]. Low-carbon behavior consists of three types: habitual behavior, consumer consumption behavior and resource recycling behavior, and LTB is a subset of low-carbon behavior [44]. According to the 2011 China Energy Development Report, LTB refers to a transportation mode where passengers can reduce carbon emissions while traveling. Furthermore, LTB is considered a type of pro-environmental behavior [45], contributing to reducing the exacerbation of climate change [46]. In urban transportation, individuals can practice LTB by opting for low-carbon travel modes such as walking, cycling and using public transportation [47]. This study defines LTB as behavior in which passengers utilize low-carbon travel modes during travel.

### 2.2. The TPB Framework

TPB was proposed in 1985, and extends the Theory of Reasoned Action (TRA) to goal-directed behavior, suggesting that intention can control behavior. Building upon TRA, TPB introduces perceived behavioral control, asserting that individual behavioral intention is determined by behavioral attitude, subjective norms and perceived behavioral control [7,30]. TPB is a well-established psychological theory that provides valuable insights into understanding and predicting individuals' behavior [35,41,42]. TPB has been widely applied in research on individual pro-environmental behavior [48]. It is used to predict individuals' environmental intention [26–28] and environmental behavior [28], alternative transportation decision [29], intention and behavior related to low-carbon travel [14,28].

#### 2.2.1. Attitude (LTA)

Attitude refers to an individual's positive or negative attitude towards a certain behavior [31]. Numerous studies have shown that individuals are inclined to a engage in a behavior when they have a positive attitude [14,45,49]. Attitude is an important predictor of individual behavior [7,18]. This study defines LTA as an individual's positive or negative attitude regarding their low-carbon travel experience. When individuals have positive LTA, they are more willing to participate in LTB. Hence, this study presents the following hypotheses:

**Hypothesis 1 (H1).** *LTA is positively related to low-carbon travel behavioral intention (LTI).*

#### 2.2.2. Subjective Norm (LTSN)

Subjective norm refers to the social pressure that individuals perceive to participate or not in a certain behavior [31]. Many studies have shown that, when an individual feels a greater social pressure, they will have stronger behavioral intention [50]. Several studies have found that subjective norms are an important predictor of behavioral intention [48,51]. This study defines LTSN as the social pressure that individuals perceive to participate or not in TPB. If people have a positive LTSN, they will feel more social pressure and are consequently more intent to engage. Hence, this study presents the following hypotheses:

**Hypothesis 2 (H2).** *LTSN is positively related to LTI.*

### 2.2.3. Perceived Behavioral Control (LTPB)

Perceived behavioral control refers to the degree of difficulty an individual perceives in a certain behavior [41]. When the individual's ability and opportunity to perform a certain behavior are greater, the individual will be more willing to perform that behavior [52]. Based on TPB and VBN, Liu et al. [7] verified that residents' attitude, subjective norms and perceived behavioral control can positively predict LTI. This study defines LTPB as an individual's perceived opportunity and ability to perform LTB. When individuals perceive that they have greater opportunity and ability, they are more willing to perform LTB. Therefore, this study proposes the following hypothesis:

**Hypothesis 3 (H3).** *LTPB is positively related to LTB.*

### 2.2.4. LTI and LTB

It is extensively acknowledged that behavioral intention is a main determinant of actual behavior [31]. Many studies have verified the predictive impact of behavioral intention on behavior [53]. For instance, tourists' pro-environmental intention accurately predicts their behavior [48]. When individuals have positive LTI, they will perform an actual behavior. Hence, the following hypothesis is proposed:

**Hypothesis 4 (H4).** *LTI is positively related to LTB.*

### 2.2.5. Mediating Effect of LTI

If individuals experience a positive attitude, subjective norms and perceived behavioral control toward a particular behavior, their behavioral intention will further affect actual behavior [31]. Behavioral intention is a primary emphasis in the TPB model [54], and works as a mediator between three predictors and actual behavior. Individuals' LTA, LTSN and LTPB can affect LTB through LTI. Hence, this study presents the following hypotheses:

**Hypothesis 5 (H5).** *LTI mediates the relationship between LTA and LTB.*

**Hypothesis 6 (H6).** *LTI mediates the relationship between LTSN and LTB.*

**Hypothesis 7 (H7).** *LTI mediates the relationship between LTPB and LTB.*

### 2.3. Moderating Effect of SF

TPB focus on the impact of individual psychological factors on behavior. However, this model ignores the external influence of contextual factors on individual behavior [36]. The transformation of behavioral intention into behavior requires stimulation, and SF is a crucial factor affecting the relationship between LTI and LTB [30].

SF is composed of multiple aspects, including economic costs, social systems, social culture, policies and regulations [55]. External policies can influence individuals' preferences for low-carbon behavior [56]. As an aspect of SF, transportation policy can lead to individuals' low-carbon travel preferences [7] Low-carbon travel policies are beneficial in guiding individuals' LTB [57,58]. For those with weak or no motivation towards low-carbon behavior, transportation policies and management regulations are more effective [59]. Furthermore, social culture significantly influences behavioral regulation. Cultural values can influence low-carbon behavior [60]. Therefore, this study expects that SF moderates the relationship between LTI and LTB and presents the following hypothesis:

**Hypothesis 8 (H8).** *SF moderates the relationship between LTI and LTB.*

### 2.4. Comparison of Individuals' LTB in Daily Residence and Tourism Destination

Studies have shown that 40% of daily behaviors are habitual [61]. Pro-environmental behavior in daily activities is habitual and rarely deliberate, and it is beneficial for individuals to engage in pro-environmental behavior continuously [48,62]. Hence, it is easier for individuals to maintain LTB in their daily residences. On the contrary, tourism provides a permissive environment for tourists [63]. In the tourism context, individuals tend to prioritize personal pleasure over environmental responsibility [64–66]. A study conducted by tourism geographers suggests that individuals who engage in environmentally friendly behaviors within their homes and immediate surroundings often find it challenging to transfer these eco-conscious practices when entering a tourist environment. Those who appear highly dedicated to environmental actions still tend to maintain their flights while traveling [6]. Individuals feel a sense of freedom and often disregard the daily norms and customary rules in tourism environment [67–69]. Individuals' behavior in the tourism environment differs from that of the daily environment, and the tourism environment can lead individuals disregardomg social norms [70]. Therefore, this study proposes the following hypothesis:

**Hypothesis 9 (H9).** *There is a huge LTB gap between daily residence and tourism destination.*

Figure 1 displays the hypothetical model.

**Figure 1.** Hypothetical model.

### 3. Methodology

#### 3.1. Questionnaire Design

Three sections made up the questionnaire. The first part aims to collect the respondents' demographic information, gender, age, income, education, fixed family size and whether or not they own a car, are gathered. The second part aims to obtain information on the individuals' LTB in their daily residence, and measures LTA, LSN, LTPB, LTI, LTB and SF, respectively. The third part aims to obtain information on the individuals' LTB during their last tourism trip, and measures LTA, LSN, LTPB, LTI, LTB and SF, respectively. SF is an extension of TPB in this study. All items are adapted from mature scales and are listed in Table 1.

**Table 1.** Survey items.

| Latent Variable | | Measurement Item | Sources |
|---|---|---|---|
| LTA | LTA1 | Low-carbon travel makes me happy in daily residence. | [7,14] |
| | LTA2 | Low-carbon travel makes me comfortable in daily residence. | |
| | LTA3 | Low-carbon travel is convenient in daily residence. | |
| | LTA4 | I think low-carbon travel is meaningful in daily residence. | |
| | LTA5 | When I travel in daily residence, I will consider whether the way to travel is low-carbon. | |
| | LTA6 | I think low-carbon travel can solve the problem of environmental pollution in daily residence. | |
| LTSN | LTSN1 | My family encourages me to choose low-carbon travel modes in daily residence. | [7] |
| | LTSN2 | Colleagues and friends encourage me to choose low-carbon travel modes in daily residence. | |
| | LTSN3 | The news media encourages me to choose low-carbon travel modes in daily residence. | |
| LTPB | LTPB1 | Public transportation can meet the needs of my daily travel completely. | [7,14] |
| | LTPB2 | I am sure I will prefer to take low carbon travel modes in daily residence next year. | |
| | LTPB3 | I believe I will adopt low-carbon travel even if others say it is not that important in daily residence. | |
| | LTPB4 | I can choose low-carbon travel as long as I want to in daily residence. | |
| LTI | LTI1 | I strongly intend to take low-carbon travel next year in daily residence. | [7] |
| | LTI2 | As far as possible, I will choose low-carbon travel in daily residence next year. | |
| | LTI3 | I will encourage the members of family and friends to choose low-carbon travel modes in daily residence. | |
| SF | SF1 | Low-carbon behavior if required by environmental laws and regulations in daily residence. | [55] |
| | SF2 | If there are policy rewards in daily residence, low-carbon behavior will be carried out. | |
| | SF3 | Social media has an impact on low-carbon behavior in daily residence. | |
| LTB | LTB1 | I always participate low-carbon travel behavior in daily residence. | [21,49] |
| | LTB2 | I talk with friends about problems related with low-carbon. | |
| | LTB3 | I prefer low-carbon travel in my daily life. | |
| | LTB4 | In the past month, the percentage of low-carbon transport options you used in daily residence. | |

This study constructs observable variables and ensures the questionnaire's validity. To ensure the accuracy and equivalence of the translation, this study utilizes classic and reverse translation techniques. Many studies have adopted the five-point Likert scale [34]. The five-point Likert scale is more convenient for respondents to select the matched option. This study uses the five-point Likert scale (1 means completely disagree, 5 means completely agree) to measure all items.

*3.2. Data Collection*

Prior to the official distribution of the questionnaires, 120 pre-test questionnaires were administered to the respondents through the Internet to assess the validity of questionnaire design. We made modifications to the questionnaire, addressing any misunderstandings and ambiguities identified during the pre-test phase, in order to enhance its comprehensiveness. The formal survey was carried out online between February 2023 and March 2023 using Credamo, a widely-used online survey platform in China. Credamo has been applied in multiple previous studies [71,72] for distributing and collecting questionnaires. This study used the paid sample service provided by Credamo. Credamo's paid sample service includes over 3 million participants from various areas and demographic backgrounds in China. For this study, 550 questionnaires were distributed and 506 valid questionnaires were acquired after removing invalid responses and responses finished within 2 min, with a total effective response rate of 92.0%. The survey was ethically conducted and all respondents participated voluntarily.

*3.3. Research Models*

In order to investigate the complicated interrelationship among individual's LTB and other variables under the two contexts of daily residence and tourism destination,

and to confirm that the data and hypothetical model is suitable for further analysis, this study uses SPSS 22.0 for descriptive statistical analysis, reliability and validity testing. Afterwards, PLS-SEM is conducted using Mplus 8.0 to estimate the relationship among variables, including the direct effect, mediating effect and moderating effect. PLS-SEM estimates partial model structures by combining principal component analysis and ordinary least squares regression. In contrast, CB-SEM relies on the data's covariance matrix and estimates model parameters by considering shared variance [73]. PLS-SEM, on the other hand, is a variance-based SEM technique [73]. PLS allows researchers to estimate complex models without imposing distribution assumptions on the data [74] and is specifically used in an exploratory manner for testing path model hypotheses [75]. It has been widely applied in fields such as tourism, marketing, and management research [27,76].

SPSS 22.0 was used to examine pre-test data. The Cronbach's α of the whole scale of the questionnaire is 0.873 > 0.8, and each variable is greater than 0.7, which indicates that the scale has good inner consistency and reliability. Meanwhile, KMO = 0.821 > 0.7 and Bartlett's vales of $p < 0.001$ show that the scale has good validity, making it a good candidate for factor analysis. Thus, the scale can be applied to the formal survey.

## 4. Results

### 4.1. Descriptive Statistical Analysis

The sample in this study is well balanced, with 56.3% (n = 285) male and 43.7% (n = 221) female. In terms of age, five groups are mentioned in the questionnaire (18–25, 26–35, 36–45, 46–60, >60), and 32.2% (n = 163) have an age of 36–45. In terms of education level, four groups are mentioned in the questionnaire (junior high school/high school, junior college, undergraduate, master and above), and 42.7% (n = 216) have a bachelor's degree. In terms of monthly income, five groups are mentioned in the questionnaire (≤2000, 2001–4000, 4001–6000, 6001–8000, >8000), and for most of respondents this is more than RMB 4001–6000 (23.1%). In terms of family fixed population, four groups are mentioned in the questionnaire (1, 2, 3, ≥ 4), and 33.8% (n = 171) have 2 people in their family. Among the respondents, 49.6% (n = 251) own a private car. The demographic characteristics are shown in Table 2.

**Table 2.** Demographic characteristics.

| Feature | Type | Frequency | Percentage/% |
|---|---|---|---|
| Gender | Female | 221 | 43.7 |
| | Male | 285 | 56.3 |
| Age | 18–25 | 63 | 12.5 |
| | 26–35 | 160 | 31.6 |
| | 36–45 | 163 | 32.2 |
| | 46–60 | 120 | 23.7 |
| | >60 | 0 | 0 |
| Education level | Junior high school/high school | 57 | 11.3 |
| | Junior college | 156 | 30.8 |
| | Undergraduate | 216 | 42.7 |
| | Master and above | 77 | 12.7 |
| Income | ≤2000 | 83 | 16.4 |
| | 2001–4000 | 110 | 21.7 |
| | 4001–6000 | 117 | 23.1 |
| | 6001–8000 | 93 | 18.4 |
| | >8000 | 103 | 20.4 |
| Population | 1 | 82 | 16.2 |
| | 2 | 171 | 33.8 |
| | 3 | 112 | 22.1 |
| | ≥4 | 141 | 27.9 |
| Private car | Yes | 251 | 49.6 |
| | No | 255 | 50.4 |

### 4.2. Reliability and Validity Test

Two model revisions were performed during the confirmatory factor analysis procedure, resulting in the removal of two items with a factor loading below 0.6. Subsequently, tests were conducted to assess reliability, validity, and the fit of the structural model.

#### 4.2.1. Reliability and Convergent Validity Test

Reliability testing is to test whether a group of items in the questionnaire can reflect the same concept, that is, internal consistency. When KMO > 0.7 and Bartlett $p \leq 0.01$, which are applied to assess correlation and independence between variables, this indicates that the questionnaire is appropriate [77].

In this study, KMO = 0.884, and Bartlett $p < 0.001$. Furthermore, we measured the factor loading of each item (as shown in Table 3). It is considered to be ideal when the factor loading is greater than 0.6 [78]. The Composite Reliability (CR) coefficient of the main variables are all greater than 0.8, showing that the internal consistency of the items is good. The Average Variance Extracted (AVE) of the main variables are all greater than 0.5, indicating that the variables have good convergent validity.

**Table 3.** Testing of validity and reliability and VIF.

| Feature | Item | Estimate | S.E. | Est./S.E. | $p$-Value | R-Square | CR | AVE | VIF |
|---------|------|----------|------|-----------|-----------|----------|-----|-----|-----|
| LTA | LTA1 | 0.732 | 0.024 | 31.020 | 0.000 | 0.536 | 0.891 | 0.576 | 2.094 |
| | LTA2 | 0.848 | 0.016 | 52.310 | 0.000 | 0.719 | | | 2.916 |
| | LTA3 | 0.748 | 0.023 | 33.261 | 0.000 | 0.560 | | | 2.122 |
| | LTA4 | 0.712 | 0.025 | 28.767 | 0.000 | 0.507 | | | 1.945 |
| | LTA5 | 0.744 | 0.023 | 32.711 | 0.000 | 0.554 | | | 2.152 |
| | LTA6 | 0.764 | 0.022 | 35.547 | 0.000 | 0.584 | | | 2.227 |
| LTSN | LTSN1 | 0.778 | 0.026 | 29.844 | 0.000 | 0.605 | 0.810 | 0.587 | 2.005 |
| | LTSN2 | 0.776 | 0.026 | 30.193 | 0.000 | 0.602 | | | 1.995 |
| | LTSN3 | 0.745 | 0.027 | 27.585 | 0.000 | 0.555 | | | 1.948 |
| LTPB | LTPB1 | 0.753 | 0.026 | 28.446 | 0.000 | 0.567 | 0.818 | 0.530 | 2.052 |
| | LTPB2 | 0.758 | 0.026 | 29.057 | 0.000 | 0.575 | | | 1.928 |
| | LTPB3 | 0.702 | 0.029 | 24.323 | 0.000 | 0.493 | | | 1.761 |
| | LTPB4 | 0.697 | 0.029 | 24.130 | 0.000 | 0.486 | | | 1.755 |
| LTI | LTI1 | 0.787 | 0.023 | 34.921 | 0.000 | 0.619 | 0.828 | 0.616 | 2.179 |
| | LTI2 | 0.759 | 0.024 | 31.411 | 0.000 | 0.576 | | | 2.039 |
| | LTI3 | 0.807 | 0.022 | 37.491 | 0.000 | 0.651 | | | 2.245 |
| SF | SF1 | 0.816 | 0.023 | 35.078 | 0.000 | 0.666 | 0.837 | 0.631 | 2.179 |
| | SF2 | 0.781 | 0.025 | 31.807 | 0.000 | 0.610 | | | 2.213 |
| | SF3 | 0.785 | 0.024 | 32.203 | 0.000 | 0.616 | | | 2.026 |
| LTB | LTB1 | 0.676 | 0.029 | 23.411 | 0.000 | 0.457 | 0.829 | 0.549 | 1.723 |
| | LTB2 | 0.751 | 0.025 | 30.299 | 0.000 | 0.564 | | | 1.957 |
| | LTB3 | 0.748 | 0.025 | 30.060 | 0.000 | 0.560 | | | 1.963 |
| | LTB4 | 0.785 | 0.023 | 34.143 | 0.000 | 0.616 | | | 2.173 |

#### 4.2.2. Discriminant Validity Test

The square root of the AVE value for each variable, as well as the correlation coefficients between variables, are always utilized to assess discriminant validity. All variables have a positive correlation (see Table 4). Furthermore, the square root of the AVE for each observed variable exceeds its correlation coefficient with other variables, providing evidence of the scale's strong discriminant validity [79].

**Table 4.** Discriminant Validity Test.

| Variables | LTA | LTSN | LTPB | LTI | SF | LTB |
|---|---|---|---|---|---|---|
| LTA | 0.759 | | | | | |
| LTSN | 0.315 | 0.766 | | | | |
| LTPB | 0.266 | 0.284 | 0.728 | | | |
| LTI | 0.519 | 0.611 | 0.392 | 0.785 | | |
| SF | 0.128 | 0.183 | 0.268 | 0.198 | 0.794 | |
| LTB | 0.502 | 0.438 | 0.306 | 0.644 | 0.276 | 0.741 |

Hair et al. [80] recommended assessing discriminant validity by examining the HTMT (heterotrait–monotrait ratio of correlations) values. In this method, discriminant validity is considered satisfactory when the HTMT value remains below the threshold of 0.9. Utilizing smart-PLS, Table 5 further demonstrates that the highest HTMT value is 0.644 (LTI-LTB), indicating good discriminant validity among the latent variables.

**Table 5.** HTMT values.

| Variables | LTA | LTSN | LTPB | LTI | SF | LTB |
|---|---|---|---|---|---|---|
| LTA | | | | | | |
| LTSN | 0.319 | | | | | |
| LTPB | 0.263 | 0.287 | | | | |
| LTI | 0.528 | 0.613 | | | | |
| SF | 0.126 | 0.188 | 0.264 | 0.196 | | |
| LTB | 0.515 | 0.438 | 0.309 | 0.644 | 0.287 | |

### 4.2.3. Multicollinearity Test

The Variance Inflation Factor (VIF) is commonly applied to assess multicollinearity between scales. In accordance with the standard of multicollinearity test, a VIF < 10 indicates the absence of multicollinearity among the variables [81]. As shown in Table 3, all observed variables' VIFs < 10, which demonstrates that there is no significant multicollinearity. The maximum score is 2.916 and the average score is 2.069. These results prove that there is no serious multicollinearity among the variables, enabling a further analysis to be carried out.

### 4.2.4. Structural Model Fit Test

The structural model fit test is to evaluate the fitting results of the model and Mplus 8.0 is utilized to evaluate the fit of the structural model in two contexts.

A series of structural model fit tests with a maximum likelihood estimation in Mplus 8.0 are conducted to examine whether our focal variables are distinctive constructs in two contexts (see Table 6). The results reveal that the six-factor model in daily residence provides a satisfactory fit to the data ($\chi^2$/df = 2.082, SRMR = 0.038, RMSEA = 0.046, CFI = 0.956, TLI = 0.949). Besides, model comparison results show that the hypothetical six-factor model fits the data markedly better than the other combined models. Meanwhile, the proposed six-factor model for tourism destination shows a good fit to the data.

**Table 6.** Goodness-of-fit testing of structural model.

| Model | | Absolute Model Fit | | | Incremental Model Fit | |
|---|---|---|---|---|---|---|
| | | $\chi^2$/df | SRMR | RMSEA | CFI | TLI |
| | Six-factor model | 2.082 | 0.038 | 0.046 | 0.956 | 0.949 |
| | Five-factor model | 4.513 | 0.083 | 0.083 | 0.855 | 0.833 |
| | Four-factor model | 6.747 | 0.100 | 0.107 | 0.759 | 0.728 |
| Daily residence | Three-factor model | 9.037 | 0.115 | 0.126 | 0.658 | 0.619 |
| | Two-factor model | 10.615 | 0.127 | 0.138 | 0.587 | 0.544 |
| | One-factor model | 12.596 | 0.123 | 0.151 | 0.500 | 0.450 |

**Table 6.** *Cont.*

| Model | | χ²/df | SRMR | RMSEA | CFI | TLI |
|---|---|---|---|---|---|---|
| | | **Absolute Model Fit** | | | **Incremental Model Fit** | |
| Tourism destination | Six-factor model | 1.773 | 0.035 | 0.039 | 0.967 | 0.961 |
| | Five-factor model | 4.155 | 0.104 | 0.105 | 0.753 | 0.721 |
| | Four-factor model | 6.537 | 0.095 | 0.110 | 0.723 | 0.687 |
| | Three-factor model | 8.306 | 0.118 | 0.120 | 0.670 | 0.632 |
| | Two-factor model | 11.179 | 0.139 | 0.142 | 0.534 | 0.488 |
| | One-factor model | 13.093 | 0.128 | 0.155 | 0.447 | 0.392 |
| Evaluation criterion | | 1–3 | <0.08 | <0.08 | >0.9 | >0.9 |
| Six-factor model Results | | Good | Good | Good | Good | Good |

*4.3. Hypotheses Analysis*

4.3.1. Direct Path Effects Analysis

SEM is always applied to examine the significance of the path coefficients of two contexts models (see Table 7 and Figure 2). Apparently, for the daily residence model, LTA has a markedly positive effect on LTI (bH1 = 0.328, $p < 0.001$), LTSN has a markedly positive effect on LTI (bH2 = 0.459, $p < 0.001$), LTPB has a significantly positive effect on LTI (bH3 = 0.174, $p < 0.01$), and LTI also has a markedly positive effect on LTB (bH4 = 0.467, $p < 0.001$). Hence, H1, H2, H3 and H4 are supported for daily residence. For the tourism destination model, except for H2 and H3, H1 and H4 are supported. All in all, the direct paths effects in the two contexts are different. Meanwhile, the path coefficient from LTI to LTB is greater in daily residence than tourism destination. H9 is supported.

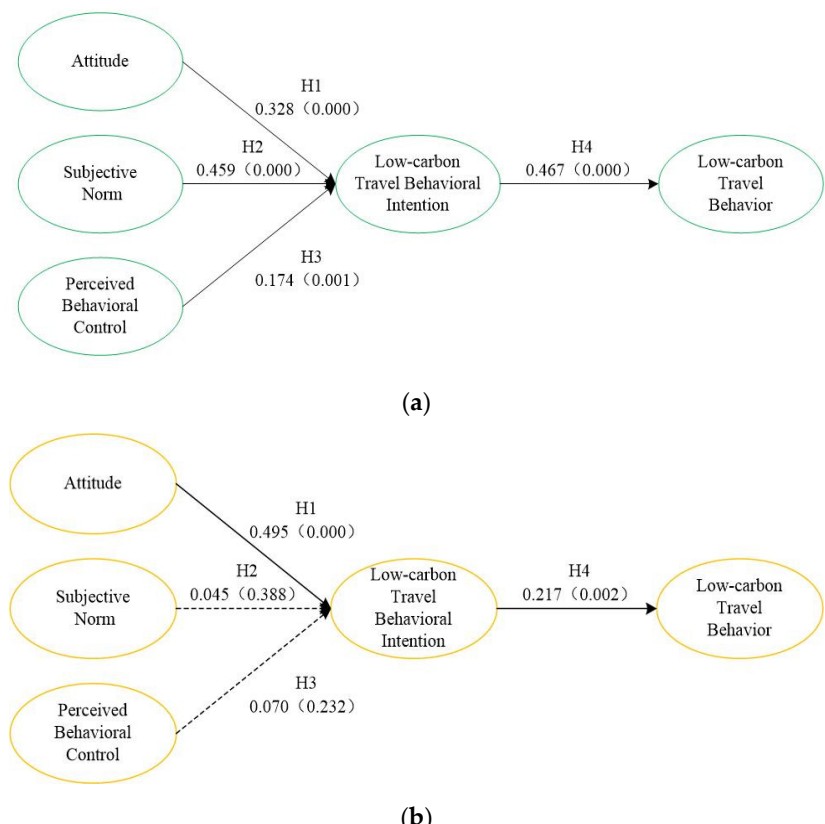

(**a**)

(**b**)

**Figure 2.** Direct paths effects test. (**a**) Direct paths effects effect of daily residence model; (**b**) Direct paths effects effect of tourism destination model.

**Table 7.** Direct paths effects test.

| | Hypotheses Tested | | Estimate (*p*) | SE | Est./S.E. | Results |
|---|---|---|---|---|---|---|
| Daily residence | | | | | | |
| H1 | LTA→LTI | | 0.328 *** | 0.046 | 7.098 | Support |
| H2 | LTSN→LTI | | 0.459 *** | 0.048 | 9.546 | Support |
| H3 | LTPB→LTI | | 0.174 ** | 0.050 | 3.480 | Support |
| H4 | LTI→LTB | | 0.467 *** | 0.071 | 6.618 | Support |
| Tourism destination | | | | | | |
| H1 | LTA→LTI | | 0.495 *** | 0.059 | 8.445 | Support |
| H2 | LTSN→LTI | | 0.045 | 0.052 | 0.864 | Not Support |
| H3 | LTPB→LTI | | 0.070 | 0.059 | 1.195 | Not Support |
| H4 | LTI→LTB | | 0.217 ** | 0.069 | 3.150 | Support |

Note: *** *p* < 0.001, ** *p* < 0.01.

#### 4.3.2. Mediating Effect Test

Compared with the Soble test, Bootstrapping is a more effective method for testing indirect effects [82]. Bootstrapping is constructed using Mplus 8.0 in this study. The mediating effects of LTI are tested by repeated sampling 5000 times and constructing a 95% confidence interval (see Table 8).

**Table 8.** Mediating effect test.

| Hypotheses Tested | | Daily Residence | | | Tourism Destination | | |
|---|---|---|---|---|---|---|---|
| | | Direct Effect | Indirect Effect | Mediating Effect | Direct Effect | Indirect Effect | Mediating Effect |
| H5 | LTA→LTI→LTB | 0.175 *** | 0.119 *** | Partial | 0.223 *** | 0.108 ** | Partial |
| H6 | LTSN→LTI→LTB | 0.041 | 0.127 *** | Full | 0.051 | 0.010 | No |
| H7 | LTPB→LTI→LTB | 0.031 | 0.056 ** | Full | 0.065 | 0.015 | No |

Note: *** *p* < 0.001, ** *p* < 0.01.

Specifically, for the daily residence model, the direct effect of LTA on LPB is significant (b = 0.175, *p* < 0.001, 95% CI = [0.120, 0.325]) and the indirect effect is significant (b = 0.119, *p* < 0.001, 95% CI = [0.094, 0.222]), which suggests that LTA has a significant and positive impact on LTB through LTI, that is, LTI plays a partial mediating effect. H5 is supported. Meanwhile, the direct effect of LTSN on LPB is not significant (b = 0.041, *p* > 0.05, 95% CI = [−0.053, 0.187]) and the indirect effect is significant (b = 0.127, *p* < 0.001, 95% CI = [0.142, 0.298]). The results demonstrate that LTSN has a significant and positive impact on LTB through LTI, which means that LTI plays a full mediating effect. H6 is supported. Likewise, the direct effect of LTPB on LPB is not significant (b = 0.031, *p* > 0.05, 95% CI = [−0.058, 0.146]), while the indirect effect is significant (b = 0.056, *p* < 0.01, 95% CI = [0.034, 0.140]). The results show that LTI plays a full mediating effect. H7 is supported. Meanwhile, for the tourism destination model, H5 is supported, but H6 and H7 are rejected.

Overall, the mediating effect of LTI (H5–H7) is more significant in the daily residence model. For the daily residence model, three mediating effect paths are valid. For the tourism destination model, only the LTA-LTI-LTB path is valid. The mechanism proposed by TPB has not been able to be proved in the tourism destination model. Hence, there are differences in LTB between daily residence model and tourism destination model. H9 is supported.

#### 4.3.3. Moderating Effect Test

We used Mplus 8.0 to test the moderating effect of SF in daily residence and tourism destination models (see Table 9). Specifically, for the daily residence model, results show that the interaction effect between LTI and SF on LTB is significant (b = 0.313, *p* < 0.001). To better describe the moderating effect of SF, we test the significance of simple slopes at high and low levels of SF. Results show that the moderating effect of SF is significant both when SF is at a level (b = 0.763, *p* < 0.001) and at a low level (b = 0.190, *p* < 0.001). H8 is supported.

**Table 9.** Moderating effect test.

| Daily Residence | | | | | Tourism Destination | | | | |
|---|---|---|---|---|---|---|---|---|---|
| **SF** | **Effect** | **SE** | **t** | ***p*** | **SF** | **Effect** | **SE** | **t** | ***p*** |
| Int | 0.313 | 0.038 | 8.333 | 0.000 | Int | 0.556 | 0.050 | 11.102 | 0.000 |
| Low | 0.190 | 0.049 | 3.852 | 0.000 | Low | −0.081 | 0.055 | −1.470 | 0.142 |
| High | 0.763 | 0.051 | 15.065 | 0.000 | High | 0.736 | 0.061 | 12.140 | 0.000 |

For the tourism destination model, results show that the interaction effect between LTI and SF on LTB is significant (b = 0.556, $p < 0.001$). Likewise, the moderating effect of SF is significant when SF is at a high level (b = 0.736, $p < 0.001$). Conversely, the moderating effect of SF is not significant when SF is at a low level (b = −0.081, $p > 0.05$). The moderating effect is painted in Figure 3.

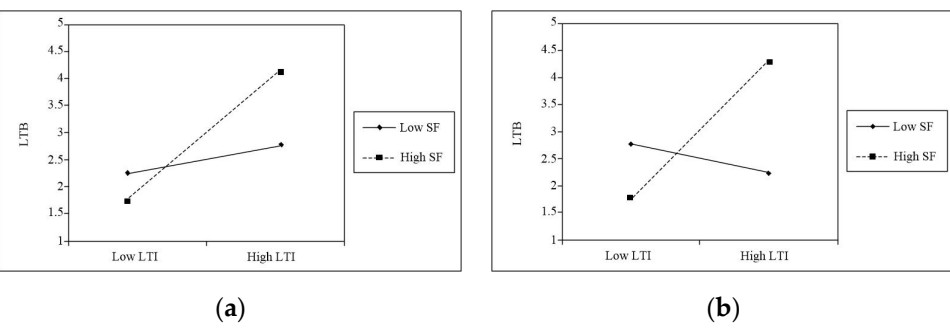

(a)　　　　　　　　　　　　　　　　　　　　　　　　　(b)

**Figure 3.** Interactive effect of LTI and SF on LTB in daily residence model (**a**) and tourism destination model (**b**).

All in all, the overall moderating effect slope is greater for tourism destination than daily residence (0.556 > 0.313). For the daily residence model, the moderating effect of SF is significant both at high and low levels. For the tourism destination model, the positive moderating effect is significant only when SF is at a high level. These results show that the moderating effect of SF shows differences between the two models. H9 is supported.

## 5. Discussion

This study demonstrates originality by integrating TPB and ABC models to examine the disparities between individuals' LTB in daily residence and tourism destination.

### 5.1. Comparison between Daily Residence and Tourism Destination

The findings in Section 4.3.1 show that the TPB-ABC model has a greater explanatory and predictive capability for daily residence than tourism destination. To some extent, this indicates that different contexts influence individuals' LTB. Specifically, in the context of daily residence, LTSN has a significant impact on the LTI. In the tourism destination context, the impact of LTSN on LTI is not significant, displaying the opposite mechanism proposed in TPB. However, this finding is in contradiction with the findings of Li et al. [83], who found that there is a significant positive correlation between residents' LTSN and LTI. This result has certain practical rationality. Individuals choose transportation modes in daily life from force of habit. In tourism destination, individuals ignore the daily norms and behavior rules [69], making it more difficult for individuals' LTSN to trigger LTI. This relationship was also not confirmed in the studies of López-Mosquera et al. [84] and Hu et al. [14]. Secondly, in tourism destination, LTPB has no significant impact on LTI, which is contrary to the context of daily residence. This is different from some previous studies, which found that there is a significant positive correlation between residents' LTPB and LTI [50]. Individuals will reduce their control over the physical and social environment when in an unfamiliar environment, which will influence their behavior [81]. Hence, this result has

certain practical rationality. Thirdly, there is also a certain gap between daily residence and tourism destination, in the path from LTI to LTB. In the context of tourism destination, individuals' LTI has a significant and positive effect on LTB. However, compared with daily residence, this effect is weaker. This result is consistent with existing research to some content [85]. In tourism contexts, there are differences between individuals' pro-environmental intention and behavior [86]. Thus, for tourism destination, the influence of individual's LTI on LTB is weakened.

### 5.2. Mediating Effect of LTI

This study indicates that LTI plays a significant mediating role, especially for daily residence. Firstly, LTI plays a significant mediating role between LTA and LTB in both contexts, which is in line with the mechanism suggested in TPB theory. An individual with positive LTA usually intends to choose low-carbon transportation modes and takes action. The findings of this study confirm the importance of LTA for LTI and LTB, which is consistent with the findings of Jia et al. [45]. Secondly, in daily residence, LTI plays a significant mediating role between LTSN and LTB. However, the mediating role of LTI for tourism destination is not significant, which is contrary to previous studies [49]. The reason for this may be that the loose tourism environment makes individuals avoid norms [70]. That is, for tourism destination, the LTSN perceived by individuals cannot affect LTB through LTI.

Thirdly, for daily residence, LTI plays a significant mediating role between LTPB and LTB. Contrary to our hypothesis, the mediating effect of LTI is not significant. It has been suggested that perceived behavioral control represents not only perceived physical control, but also perceived behavioral difficulty and social appropriateness [87]. Individuals tend to reduce their control over the physical and social environment when in an unfamiliar tourism environment, which has an impact on behavior [88,89]. For example, it is difficult for people to choose low-carbon transportation modes when there are no buses and shared bicycles nearby. These issues restrict the ability of individuals to select low-carbon transportation modes in a tourism environment. This possibility can explain why, for tourism destination, the LTPB–LTI–LTB path is not significant.

### 5.3. Moderating Effect of SF

According to the results in Section 4.3.3, we can see that SF has a positive moderating effect on the relationship between LTI and LTB in the two contexts. These findings illustrate that individual behavior is the result of the combined effects of psychological factors and contextual factors [90–92].

Individuals' LTB is not only affected by LTI, but also affected by the interaction of LTI and SF. Specifically, in daily residence, the positive impact of LTI on LTB is stronger when SF is at a high level. This demonstrates that laws and regulations, policy incentives and social media in daily residence can promote the transition from individual's LTI to LTB. Meanwhile, for tourism destination, although the individuals' LTI has a significant positive effect on LTB, the effect is weak. The moderating effect is not significant when SF is at a low level. When the level of SF is high, the effect of LTI on LTB is significantly enhanced. This result is consistent with the existing studies, showing that stronger laws and regulations, policies and social media related to low-carbon travel are very effective in strengthening individuals' LTB [93].

## 6. Conclusions

### 6.1. Theoretical Implications

Based on TPB and ABC, this paper analyses the differences of individuals' LTB in the context of daily residence and tourism destination and explore its internal realization path and mechanism. This study has three theoretical implications: (1) This study uses the TPB-ABC integrated model to explore the individual's LTB. The results show that the TPB-ABC integrated model provides better explanation and prediction ability for different

contexts of daily residence and tourism destination. This study is a further expansion of TPB theory and ABC theory. (2) This study explores the internal realization path and influence mechanism of individual LTB based on the TPB-ABC integration model. Results show that LTI plays a significant mediating role for both daily residence and tourism destination, especially for daily residence. SF strengthens the influence of LTI on LTB, especially for tourism destination. Passengers' LTB is an important factor influencing low-carbon transportation [14]. This study promotes research on LTB. (3) This study distinguishes the differences between LTB in different contexts. Results show that there is a certain gap in the individuals' LTB between daily residence and tourism destination. Previous studies distinguish differences in individuals' travel behavior in daily life and tourism environments, which focused on travel behavior between cities rather than intra-destination [17]. Furthermore, some studies have used multiple samples to explore the differences between individuals' LTB in different contexts [24], which were limitations of the research. This study bridges the gaps in existing research and is an in-depth piece of research on low-carbon travel in different contexts.

*6.2. Managerial Implications*

This study aims to explore the internal mechanism of individuals' LTB and the differences between daily life and tourism destination. Travel behavior is the key to analyzing and managing low-carbon transportation in cities [45]. The results show that, compared with daily residence, the individuals' LTB in tourism destinations is weakened. This study contributes to a deeper understanding of individuals' LTB and provides practical reference value for promoting low-carbon travel in the whole of society. This study provides an opportunity for public policy makers and tourism destinations to formulate some interventions and has certain managerial implications.

Tourism destinations should promote individuals' LTB through the following aspects: (1) Enhance the atmosphere guidance of individuals' LTSN. The relaxed environment of tourism destinations makes it easy for individuals to ignore norms. Strengthening LTSN could promote individuals to pay attention to the rules of low-carbon travel, thereby conducting individuals' LTB, for example, increase publicity on environmental issues and low-carbon travel in public places and online ticketing systems. (2) Strengthen the construction of low-carbon travel infrastructure in tourism destinations. The lack of infrastructure construction limits the ability of individuals to choose low-carbon transportation modes in tourism destination. Destinations should strengthen infrastructure construction, reduce the difficulty for individuals to choose low-carbon transportation modes in tourism destinations, which will improve individual's LTI and LTB. For example, improve the construction of low-carbon transportation facilities. Destinations should carry out overall management of public transportation hubs, electric vehicle charging places and public activity places. In addition, destinations can increase the number of parking spots for shared vehicles, and plan the routes of buses and subways scientifically. At the same time, with the development of the Internet and big data, destinations should actively promote the use of emerging technologies such as mobile payment and electronic tickets in the transportation field, strengthen the monitoring of low-carbon transportation and release effective information through the Internet to guide people in low-carbon travel. (3) Bolster SF. The results show that SF contributes to transition of individuals' LTI to LTB both in daily residence and tourism destination. Therefore, the government and relevant departments should promote supportive policies and manage regulations of low-carbon travel, strengthen low-carbon publicity and guidance through social media. For example, carry out low-carbon online activities, popularize low-carbon knowledge and policies to make people more aware of low-carbon behaviors and enhance LTI and LTB.

*6.3. Limitations*

Furthermore, this study also has some limitations. Firstly, based on the questionnaire data, this study explores the individual's LTB and analyzes the differences between

daily residence and tourism destination. However, this study does not analyze whether the two contexts of LTB have some relationships. In the future, we can investigate the intrinsic relationship between the LTB in different contexts. Secondly, this study explores the moderating effect of SF, emphasizing the influence of low-carbon travel policies and social media. Previous studies have shown people's pro-environmental behavior vary from different economic conditions and cultures [94–96]. Urban and rural areas are an important format in comparative contextual researches [35]. There may be differences of the individuals' LTB in urban and rural areas. In the future, the contexts of urban and rural tourism destinations can be added to explore the LTB. In addition, to maintain a broader perspective and generalize our findings to a wider range of tourist destinations, this study employed a comprehensive online survey and did not conduct on-site research. This constitutes a limitation of this paper. In future research, we will consider investigating specific destinations and use longitudinal studies to explore the differences in tourists' low-carbon travel behaviors between daily residence and particular tourist destination.

**Author Contributions:** L.W.: Conceptualization, Formal analysis, Funding acquisition, Investigation, Methodology, Project administration, Supervision, Writing—original draft, Writing—review & editing. J.W.: Investigation, Writing—review & editing. P.S.: Investigation, Writing—review & editing. S.L.: Methodology, Validation. S.Z.: Investigation, Writing—original draft. All authors have read and agreed to the published version of the manuscript.

**Funding:** This research was funded by the Fundamental Research Funds for the Central Universities (No. 2022YJS053), Fundamental Research Funds for the Central Universities (No. B19JBW200040) and Fundamental Research Funds for the Central Universities (No. 2019JBWB002).

**Institutional Review Board Statement:** Not applicable.

**Informed Consent Statement:** Informed consent was obtained from all subjects involved in the study.

**Data Availability Statement:** No supplementary data. The data presented in this study are available on request from the corresponding author. The data are not publicly available because the data in this article were obtained through a questionnaire survey, this study was supported by the respondents, but at the same time, the authors promised that respondents' information would not be made public.

**Conflicts of Interest:** The authors declare no conflict of interest.

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
