# Peer review of "Low-Carbon Travel Behavior in Daily Residence and Tourism Destination: Based on TPB-ABC Integrated Model"

_sustainability, doi:10.3390/su151914349_

Round 1

Reviewer 1 Report

Dear editor

Dear authors,

Thank you to give me an opportunity to review this paper. This paper aims to explores the influencing factors and formation paths of individual’s low-carbon travel behavior, and analyzes low-carbon travel behavior’s difference in daily residence and tourism destination.

Below you can find my comment to improve the quality of this paper.

Introduction section has been well written.

 Literature review section: Line 128-129, author need to explain what is TPB and why author use TPB? how previous study use this TPB theory?

Methodology section: table 1 and the questionnaires in missing. Author need to add the questionnaires in this section or in appendix section. We need to check the questionnaires; however, the questionnaires will be a valuable information for readers.

The analysis data is missing. How author use MPLUS need to be explain in this section. Besides, author need to explain about CB-SEM and PLS-SEM.

Result section: line 251 descriptive statistics section need to added a table for clearer.

please provide HTMT and fornel-lacker for discriminant validity. please see this paper for example: T. T. Wijaya, Y. Cao, M. Bernard, I. F. Rahmadi, Z. Lavicza, and H. D. Surjono, “Factors influencing microgame adoption among secondary school mathematics teachers supported by structural equation modelling-based research,” Front. Psychol., vol. 13, no. September, pp. 1–16, 2022, doi: 10.3389/fpsyg.2022.952549.

References section.  

references in this paper are lacking. author can add at least 20-30 more references. a good paper requires a strong reference source

Reviewer 2 Report

Dear Author(s),

Thank you for this opportunity to read your work and take part into this review process. The article is interesting and the theme is really actual. Anyway, it is possible to improve some important aspects of your work. Below you can find the suggestions that you need to answer before publication.

Line 41: The bibliography reported is basically adequate, but it is important to quote even more recent references.

Line 44: It could be interesting to know which kind of political-economical measure China is adopting to reduce the number of vehicles: please, provide some reference about that or briefly explain this policy by providing some examples.

Line 70: Eliminate the double dot. Line 180: Eliminate double coma. 

Line 103: The text quotes "Previous studies", but the references are not present: only one study is cited. It is necessary to improve this point. Something similar happens even for line 105.

Lines 194-199: Please, provide more references: the theme of "permissive environment" in the case of tourists and "prioritize personal pleasure over environmental responsibility" appear fundamental for your work. Only 3 references are a few numbers. Moreover, please, consider that cultural geographers, sociologists, and anthropologists dedicated many works to this theme.

Results, Discussion, and Conclusions: One of the central points of this study is the way to use transport in the tourist destination, but curiously any "real" destination is taken into account. Otherwise, the context is understood in other, more abstract terms. Please, try to clarify "why" you decided to avoid a discussion or a collection of data in a real destination. The point is very interesting, but it deserves a better explanation (e.g. It could be a point to discuss in the "Limitations" section.

Round 2

Reviewer 1 Report

Dear authors,

i have read your paper and find that this paper has been well improved.

but, please avoid self-citations more than 2 papers.

the self-citations is not good for journal reputations.

all the best 

Reviewer 2 Report

Dear Author(s),

Thank you for updating your paper according to the suggestions. Despite this, my perplexity continues regarding a part of your theoretical approach: it is not possible to build a universally valid model, in the abstract, for tourism cases. On the contrary, it is necessary to contextualize the research field.

Please, proofread once again your article: some formal errors are continuing to weigh down the text (double dot, comas, etc.). 

The paper can be accepted after minor revisions (proofreading of the text).
